# Exploring the Interplay of Leisure Freedom, Satisfaction, and Participation Styles Among Turkish Sports Sciences Students

**DOI:** 10.3390/bs15091273

**Published:** 2025-09-18

**Authors:** Özden Tepeköylü-Öztürk, Mümine Soytürk

**Affiliations:** 1Faculty of Sport Sciences, Department of Recreation, Pamukkale University, 20160 Denizli, Türkiye; 2Faculty of Sport Sciences, Department of Physical Education and Sports, Manisa Celal Bayar University, 45040 Manisa, Türkiye; mumine.soyturk@cbu.edu.tr

**Keywords:** recreation, perceived freedom, participation setting, university students, leisure participation

## Abstract

Understanding the dynamics of leisure experiences is essential for promoting well-being among university students, particularly those studying sports sciences. The study aimed to explore the predictive relationship between perceived freedom in leisure and leisure satisfaction among sports sciences students. It also examined whether perceived freedom and satisfaction differ significantly by gender, leisure participation types, and activity setting (indoor vs. outdoor). To this end, a total of 3192 students from various universities participated in the research. The data were analyzed using frequency distribution, arithmetic mean, standard deviation, Pearson correlation, linear regression, three-factor ANOVA, one-way MANOVA, and (3 × 2) MANOVA. As a result, the analysis revealed significant positive relationships between perceived freedom and leisure satisfaction. Perceived freedom strongly predicted leisure satisfaction, accounting for 42% of the variance. Gender-based comparisons showed that female students reported significantly higher levels of both perceived freedom and satisfaction. Students who engaged in active, group-based leisure activities, especially in outdoor settings, experienced the highest levels of freedom and satisfaction. Overall, a ctivities that are active, social, and conducted outdoors enhance students’ perceived freedom and increase their leisure satisfaction.

## 1. Introduction

Leisure has long been recognized as a fundamental aspect of human life. In early communities, it referred to the limited time left after basic survival needs ([44]). As social structures developed, leisure began to serve broader purposes, such as preparing individuals for life through play, rituals, and ceremonies ([51]). In Ancient Greece, it carried both individual and socio-political value through art, philosophy, education, and sport ([44]; [10]; [83]). By contrast, in the Roman period, leisure was more closely associated with rest and recovery. In the Middle Ages, leisure was shaped by strict rules and rituals; however, with the Renaissance, it gained a more liberal character oriented towards entertainment and education ([44]). After the Industrial Revolution, reduced working hours and technological progress made leisure more visible in everyday life, giving it deeper psychological and social meaning at both individual and societal levels ([74]).

These historical perspectives show that the definition and significance of leisure have changed considerably over time. What was once limited to stratified social experiences is now recognized as a necessity for self-actualization and an important component of quality of life. In today’s context, leisure is not only associated with individual rest and rejuvenation but is also viewed as a factor that supports healthy lifestyles, social integration, and broader societal well-being. In this regard, the sector encompassing active and passive sport participation, health and fitness programs, management, tourism, hobbies, popular entertainment, and the arts ([62]) has emerged as a national and local economic structure, particularly under the stewardship of professions within the field of sport sciences. In addition, leisure has increasingly become supported through governmental initiatives involving funding, facilities, education, public awareness, and policy development ([44]).

The functions that leisure serves for both individuals and societies, along with its organization within institutional structures, highlight the need for a more comprehensive consideration of leisure within the scientific domain. As a result, leisure as a phenomenon, together with its structures and associated behaviors, has become the focus of many studies. In this study, we specifically examine the variable of perceived freedom, which Neulinger (1981) identified as the fundamental determinant of leisure (as cited in [79]). In addition, following [55] ([55]), who emphasized that participants pursue leisure activities for a variety of purposes and expectations, we also examine leisure satisfaction as a central factor shaping participation. The study was conducted with undergraduate students in the field of sport sciences, who, by virtue of their academic training, are closely engaged with both theoretical and practical dimensions of leisure, recreation, physical activity, and sport participation. This group is particularly relevant because their coursework and applied experiences equip them with the potential to make more conscious choices regarding the planning, management, and evaluation of leisure. Students of sport sciences are also positioned as role models in promoting healthy lifestyles, fostering active participation, and advancing effective leisure management within society. Furthermore, they represent a unique sample group, since the knowledge and experience gained during their academic training can be directly applied to professional practices related to leisure. Accordingly, examining students’ perceived freedom, levels of leisure satisfaction, and leisure behaviors (e.g., active/passive, individual/group participation, and activity setting preferences) offers insights that may guide the planning of campus recreation programs, the development of initiatives for broader population groups, and the formulation of related policies. In addition, because these students engage not only in compulsory academic activities but also in voluntary sport and recreational pursuits, they are exposed to a wide range of participation forms (active/passive, individual/group, indoor/outdoor). This makes it possible to observe more clearly the relationships among the variables examined in the study. The expectation that findings from this group could directly inform the development of university sport and recreation programs, shape leisure policies for young adults, and support public health further justified the selection of sport sciences students as the study sample. Consistent with this view, [76] ([76]) emphasized that physical education students tend to be more active during their leisure time, and that monitoring the prevalence of such active behaviors and their associated characteristics may inform institutional policies aimed at delivering programs focused on leisure practices.

Perceived freedom is regarded as a critical parameter for understanding individuals’ sense of autonomy in leisure experiences. [24] ([24]) argued that successful participation in leisure activities requires the ability to influence the initiation, process, and outcomes of such activities, which in return fosters a sense of freedom. In this study, perceived freedom in leisure is examined among students of sports sciences with respect to their gender, modes of participation (active/passive, individual/group), and activity settings. In addition, consistent with the behavioral assumption proposed by [24] ([24]), leisure experiences shaped not by others’ expectations but by individuals’ own needs and preferences strengthen the perception of freedom. The meaning of leisure is therefore context-dependent, influenced by attitudes, perceptions, and experiences at a given time and place ([91]). Among the emotions that accompany these experiences, leisure satisfaction is addressed in this research as a positive phenomenon. It is conceptualized as a multidimensional construct that varies across individuals and over time ([29]) and is influenced by both environmental and structural parameters (Manning, 2011, as cited in [18]). Although widely studied, its inclusion in this research is justified by its temporal variability and the necessity of re-examining it across diverse groups and contexts. Similarly, perceived freedom in leisure, also examined extensively in the literature, gains new relevance in this study as it is analyzed together with leisure satisfaction while considering students’ gender, participation modes (active/passive, individual/group), and activity settings. Within this framework, the study aims to provide a cross-sectional profile of Turkish university students in sports sciences and to establish a basis for future research that may develop new perspectives on leisure behavior in contemporary contexts.

A review of the literature reveals that this topic has been explored with various parameters across different groups. Research involving university students has shown that an increase in perceived freedom correlates with higher levels of life satisfaction and self-esteem ([1]). Perceived freedom is a significant factor influencing individuals’ subjective well-being ([57]). Furthermore, studies indicate that as perceived freedom grows, so do attitudes toward leisure and overall satisfaction with it ([79]). A lack of freedom during leisure activities among young people can lead to decreased motivation for engaging in those activities ([65]). When considered in relation to other factors, some studies have examined the influence of life structures, experiences, and motor skills on perceived freedom ([59]; [41]). Most of the existing literature, however, has focused on the relationships of perceived freedom with other variables and its positive effects on cognitive, emotional, and social competencies. Studies that reveal under which conditions and in what ways the perception of freedom varies are limited. Yet, perceived freedom is a critical parameter for understanding the conceptual framework of leisure and for advancing both traditional and contemporary theoretical developments in leisure studies ([91]). Therefore, in this study, perceived freedom was examined together with the variables of active/passive participation, individual/group participation, and activity settings (indoor/outdoor). Our research emphasizes new variables that may shed light on how leisure should be planned in line with changing social structures and the diverse needs and interests of young people, while also contributing to the development of practices whose benefits are already well documented in the literature.

When the literature on leisure satisfaction is reviewed, it is evident that many studies have examined its relationship with, and impact on, personal and social characteristics across different groups. Several demographic variables have been identified as influencing leisure satisfaction, including gender, age, family structure, social and personal characteristics, and environmental factors ([11]; [79]; [73]; [95]; [23]; [7]; [46]; [47]). Research with adults and university students has shown that leisure activities contribute positively to subjective well-being ([88]; [39]). Among university students, leisure satisfaction has been found to correlate positively with extraversion and negatively with emotional instability ([61]). In addition, it relates negatively to stress, thereby positively influencing how individuals cope with stressful events ([18]). Furthermore, individuals who are satisfied with their leisure tend to be happier ([61]; [82]; [88]; [39]) and calmer ([82]). One study focusing on university students highlighted that leisure satisfaction is a significant predictor of academic stress, indicating that as leisure satisfaction increases, academic stress tends to decrease ([63]). The studies mentioned above demonstrate that leisure satisfaction is closely linked not only to leisure-related variables such as motivation, perceived freedom, and participation tendencies, but also to characteristics, conditions, and skills that support individuals’ quality of life ([79]; [81]; [65]). However, research examining the specific conditions under which higher levels of satisfaction are achieved remains rather limited. For example, a study conducted in China found that activities performed in open spaces and natural settings were associated with greater leisure time satisfaction, while indoor activities did not show the same positive correlation. In fact, this study revealed that outdoor activities also had beneficial effects on depression and subjective well-being through leisure time satisfaction ([13]). Similarly, [4] ([4]) reported a strong connection between outdoor activities and leisure time satisfaction. The ongoing process of urbanization, along with people’s growing desire to connect with nature ([56]), suggests that similar findings can be observed in different societies. This finding highlights the importance of incorporating sufficient outdoor recreational spaces into urban planning, while the limited number of studies underscores the need for further research with diverse groups. In this study, leisure satisfaction was examined together with different modes of leisure participation (active/passive), preferred settings (outdoor/indoor), and forms of involvement (individual/group). The aim was to provide insights that could guide the planning of leisure opportunities, particularly those designed for young people.

Another issue addressed in this study is the comparison of leisure behaviors by gender, which has been widely examined in the literature with varying results ([79]; [63]; [61]; [95]; [67]). Differences observed between men and women in leisure interests, needs, and access to leisure opportunities across diverse groups and cultural contexts, as well as the potential influence of gender on leisure behaviors ([40]; [3]; [34]), make continued gender-based comparisons both meaningful and necessary. For this reason, gender was also included as a variable in the present study.

Building on the considerations outlined above, this study examines the relationship between perceived freedom in leisure and leisure satisfaction, with a particular focus on variables that directly shape leisure behaviors. By addressing this gap, the study highlights structural and behavioral parameters that can guide the effective planning of leisure and the development of youth-oriented recreation programs. Using a large-scale dataset collected from students of sport sciences, the study presents a comprehensive profile of young people’s perceived leisure freedom, satisfaction levels, and participation preferences within the Turkish context.

Turkey’s socio-cultural structure is one of the key factors shaping perceptions and behaviors related to leisure. Traditionally characterized by a family- and community-oriented social fabric, Turkish society tends to view leisure not only as an individual pursuit but also as a social process shared with family and friends ([33]). Within this context, young people’s leisure preferences are influenced not solely by personal interests and motivations but also by family expectations, societal values, and the broader social environment. In addition, Turkey’s cultural diversity produces variations in leisure experiences across different regions. Nature-based and community-centered activities are more prominent in rural areas, whereas urban settings tend to favor sports, entertainment, and technology-oriented pursuits ([25]; [6]).

In addition, socio-economic conditions and gender roles significantly influence young people’s leisure behaviors. For female students in particular, factors such as safety, social approval, and family support play a decisive role in shaping their participation in leisure activities ([50]). This suggests that the leisure experiences of young people in Turkey differ from the more individualistic and autonomous forms of participation observed in some other countries.

The findings of this study should be interpreted within the cultural, social, and economic context of university students in Turkey. Although students of sports sciences may appear to represent a convenient sample, young people from diverse regions bring with them distinct values and habits related to leisure. This reflects how Turkey’s multi-layered socio-cultural structure influences leisure perceptions and behaviors. Interaction among students from different cultural backgrounds facilitates the transfer and blending of leisure practices. As a result, while the findings may not be directly generalized to other cultural contexts, they provide valuable insights into the role of cultural context in shaping leisure behaviors. Such research can provide unique contributions to the development of university recreation programs and the formulation of youth-oriented leisure policies at both local and international levels.

Considering the preceding literature, the primary aim of this study is to examine the predictive relationship between perceived freedom in leisure and leisure satisfaction. In addition, the study seeks to determine whether there are differences in perceived leisure freedom and leisure satisfaction based on gender, patterns of leisure participation (active/passive, individual/group), and the setting of participation (indoor/outdoor).

### Theoretical Background

Leisure has remained significant both socially and personally over time and has been examined as a phenomenon explained by social and psychological theories. Early work that shaped present research was mostly philosophical reflections on the leisure activities of people. By the early 19th century, works of pioneer leisure theorists such as Max Kaplan and Thorstein Veblen marked the start of more empirical and analytical investigations. Sociological research initially attempted to account for leisure activities within the framework of social classes and, as referred to by [10] ([10]), subsequently evolved to concentrate on the relationship between work and leisure. Veblen’s The Theory of the Leisure Class ([87]) discussed how different classes utilize leisure, while [42] ([42]), in addressing the question “What is leisure?”, offered explanations that spanned not only environmental determinants but also inherent, individual-level determinants.

Kaplan, being a sociologist, also observed the psychological dimensions of leisure, identifying the individual-level factors that contribute to determining leisure experiences. In psychology, individual-level factors have primarily been examined in the form of research regarding individuals, particularly the parameters and nature of human experiences during leisure time ([10]). As a result, the concept of leisure has been explored not only in relation to its temporal dimension—i.e., time free from obligatory responsibilities—but also according to its culturally relative and intrinsic qualities, such as play, pleasure, intrinsic motivation, relaxation, freedom, and participation ([64]).

Here, it is obvious that leisure is not only characterized by social structures and class relations but also by individuals’ natural motivations, perceptions, and experiences. The theory has opened the door for the multidimensional characterization of the concept. A review of the literature shows that leisure is usually characterized in terms of time, activity, existence, and psychological state ([30]; [35]). In agreement with [36] ([36]), active people during leisure are typically self-directed and responsive to their own rhythms, enjoying a degree of relative autonomy to act as they please. In this framework, leisure as time is viewed as an interval free of obligation and independent of work ([43]; [85]; [30]). Similarly, “leisure as activity” is a definition of voluntary activity for enjoyment apart from work ([36]; [30]). Another school of thought holds that leisure is related to people’s attitudes and perceptions. Psychologist Neulinger (1974) explained his concept of leisure through three dimensions. He characterized leisure as non- instrumental, intrinsic (or related to intrinsic motivation), and involving perceived freedom (Neulinger 1974 cited in [43]; cited in [85]).

The idea of perceived freedom, together with external constraints, helps determine whether an activity qualifies as leisure. Neulinger defines pure leisure as participation that is both voluntary and driven by intrinsic motivation ([58]). Perceived freedom in leisure is a crucial factor for assessing leisure behavior and experiences (Kelly, 1978, cited in [91]). It is understood as a cognitive and motivational construct that reflects participants’ perceptions of the leisure activities they voluntarily choose to engage in ([2]). According to Kelly’s (1978) model, perceived freedom exists on a spectrum with low and high levels. Leisure activities can be chosen for both intrinsic enjoyment and social reasons. The freedom to choose leisure activities is subjective and often comes with certain limitations. While intrinsic and extrinsic motivations for participating in leisure activities are intertwined, engaging in leisure for intrinsic reasons suggests a higher level of perceived freedom (Kelly 1978 cited in [91]). [24] ([24]) defined freedom in leisure as the degree to which individuals are free from factors that hinder their participation in satisfying experiences. According to them, this sense of freedom encompasses not only the right to choose but also feelings of competence, perceived control, and deep involvement in activities. In this context, freedom in leisure is explained across five dimensions: perceived competence, perceived control, need satisfaction, depth of involvement, and playfulness. Perceived competence refers to the reinforcement of one’s sense of freedom through successful participation experiences, whereas perceived control pertains to the ability to influence the initiation, process, and outcomes of activities.

In this study, the definition proposed by [24] ([24]) was adopted, as it conceptualizes perceived freedom in leisure not only in terms of voluntariness or intrinsic motivation, but also through a multidimensional framework that includes competence, control, need satisfaction, depth of involvement, and playfulness. This comprehensive, experience-based perspective allows perceived freedom in leisure to be understood as more than just the right to choose. It also reflects individuals’ ability to feel competent in their chosen activities, to influence the process, and to derive genuine satisfaction from participation. At the same time, elements emphasized in the definitions of Neulinger (1974) and Kelly (1978), such as voluntariness, intrinsic motivation, and subjective perception, serve as complementary dimensions that support and enrich this perspective (Neulinger, 1974, as cited in [43]; Kelly, 1978, as cited in [91]). Accordingly, while the multidimensional definition of [24] ([24]) remains central to this study, it does not exclude other approaches that highlight the shared core elements of the concept. This integrative choice contributes to a holistic understanding of perceived freedom in leisure within the theoretical framework of the research, encompassing both its conceptual and experiential dimensions.

As another variable that can be examined together with perceived freedom when addressing leisure behavior, leisure satisfaction is an important factor that affects leisure participation and is also affected by participation. Studies show that there is a positive relationship between satisfaction and participation ([60]; [37]; [89]; [54]; [14]) and that satisfaction is an important predictor of leisure participation as well as participation tendency ([81]).

The emotions and thoughts with which individuals participate in leisure activities are important ([23]). According to Iso-Ahola, individuals learn which personal needs they can meet through leisure during early socialization. This process plays an important role in clarifying the specific activities that the individual is interested in (Iso-Ahola, 1980 cited in [79]). Therefore, leisure participants participate in leisure activities with various goals and expectations ([55]). [80] ([80]) interpret leisure satisfaction as the sense of fulfillment resulting from the benefits gained through participation in leisure activities. Similarly, [9] ([9]) define it as the satisfaction and positive feelings individuals experience from engaging in leisure activities. It is related to the extent to which people are satisfied with their leisure experiences and the fulfilment of motives, expectations, and needs (Mannell, 1989, cited in [79]). In other words, the smaller the difference between what is obtained from the leisure activity and the expectation, the more satisfaction can be achieved. While the fulfillment of compulsory needs is essential for overall life satisfaction, the sense of satisfaction derived from leisure also plays a central role ([5]). Thus, leisure satisfaction affects individuals’ perceptions of life satisfaction ([90]). Studies also show that there is a positive relationship between leisure satisfaction and life satisfaction ([72]; [11]; [82]; [95]; [53]; [16]; [45], [46]; [88]; [17]). [9] ([9]) identified six dimensions of leisure satisfaction: psychological, educational, social, relaxation, physiological, and esthetic. The psychological dimension refers to the perceived benefits derived from leisure participation. The educational dimension reflects the extent to which individuals perceive learning outcomes gained through leisure activities. Meeting new people and forming personal networks constitute the core of the social dimension. The relaxation dimension is associated with stress reduction, while the physiological dimension indicates the level of satisfaction derived from improved health conditions as a result of leisure engagement. Finally, the esthetic dimension concerns the suitability and design of the recreational environments where individuals participate in leisure activities ([9]).

Although various researchers have proposed different definitions of leisure satisfaction, most of these focus primarily on the expressive dimension of individual leisure experiences and do not offer a comprehensive conceptual framework (Iso-Ahola, 1980, cited in [79]; Mannell, 1989, cited in [79]; [80]). While such definitions make significant contributions, they often limit satisfaction to the extent of meeting individual expectations, or the subjective perception of benefits derived from participation. In contrast, the definition and model developed by [9] ([9]) make an important contribution to the literature by systematically conceptualizing leisure satisfaction across six dimensions: psychological, educational, social, relaxation, physiological, and esthetic. This multidimensional and holistic approach is particularly valuable as it encompasses not only individual emotions and perceptions but also learning, social interaction, health, and environmental factors. Furthermore, the frequent use of Beard and Ragheb’s approach in many studies underscores both its empirical validity and its widespread acceptance. For these reasons, the present study grounds the concept of leisure satisfaction in the definition and model proposed by [9] ([9]).

When considered within the scope of this study, each of the subdimensions of leisure satisfaction gains meaning in relation to perceived freedom in leisure from different perspectives. The psychological dimension pertains to individuals’ experiences of intrinsic pleasure, happiness, and stress reduction through activities they voluntarily choose. The educational dimension reflects how individuals support their learning and personal development through activities they freely engage in. The social dimension represents the degree to which individuals establish social relationships and develop a sense of belonging, depending on their modes of participation (e.g., individual or group activities). The relaxation dimension is associated with individuals’ opportunities for renewal and rest by utilizing their leisure time according to their own preferences, particularly in the face of a demanding lifestyle. Finally, the physiological dimension is linked to bodily health and energy gains, especially in activities that require active participation. The esthetic dimension highlights the influence of environmental factors on satisfaction in leisure activities carried out in indoor or outdoor settings. Accordingly, the six dimensions developed by [9] ([9]) make it possible to examine the relationship between perceived freedom in leisure and leisure satisfaction, which is the central focus of this study, within a multidimensional framework. They also provide basis for investigating the differences related to gender, modes of participation, and activity settings.

## 2. Materials and Methods

### 2.1. Research Design

This study employs a descriptive, comparative, and correlational approach to survey a large population of students in the field of sports sciences in Turkey. Its primary aim is to assess their perceived freedom in leisure and leisure satisfaction. In addition, the study compares different forms of leisure participation and evaluates the predictive relationships among various variables ([20]).

### 2.2. Participants and Procedure

The study sample includes 3192 students (*µ*_age_ = 21.87 ± 2.269), comprising 1241 females and 1951 males, studying sports science at various universities in Turkey. Multiple sampling techniques were used to ensure diversity and representativeness. The selection of universities offering sports science programs was the foundation for forming this sample. The faculties and school of physical education and sports across Turkey’s seven geographical regions were identified to assess the generalizability of the research results. Based on the number of faculties and schools of physical education and sports in each region, two to four institutions (representing at least 30%) were selected to ensure representation. This strategic selection was crucial in capturing a diverse perspective from various regions, which is vital for the robustness of our research. Following this, our objective shifted towards determining the average number of active students enrolled in the selected faculties and programs. ([19]). A total of 5500 questionnaires were then distributed to universities offering sports science education using the convenience sampling method. Of the 3427 forms voluntarily completed and returned, 3192 were included in the study. The remaining forms were excluded because they were incomplete or misleading, specifically if more than 10% of the items were left unanswered or if only one or five questions were filled in. The table of acceptable sample sizes for specific populations developed by [52] ([52]) was consulted to determine the appropriate sample size. According to this table, a sample of 384 participants is sufficient for populations of 1,000,000 or more ([52]). We have several reasons for our efforts to engage a larger population. First, given our country’s diverse geographical and sociocultural structures, we aimed to include many participants to ensure inclusiveness across the entire population. In addition, we chose to reach a broader sample due to the number of questions in the measurement tools used in the study and the various tests and variables we plan to examine. Although there is no specific standard in this regard, it is recommended to increase the sample size according to the number of parameters and the complexity of the model. It is stated that ratios such as 10:1 and 20:1 (parameters: cases) are appropriate ([49]). Accordingly, there are 49 questions in the “Personal Information Form” and other scales. It was planned to reach at least 20 times the number of questions, but more participants were reached than planned. In the end, 3192 usable forms were obtained.

Permission and support for the data collection process were obtained from the relevant faculties and schools of physical education and sports by contacting the university administrations at the Dean’s level. One or two lecturers from the institutions, who are experts in the field, provided support in obtaining data. The measurement instruments were administered to volunteer students by faculty members, who either used their own class sessions or attended those of their colleagues, following a convenience sampling approach.

The questionnaires were mailed to the schools and subsequently returned through the same postal service. Online platforms were not used for data collection because of difficulties in accessing participants’ contact information, anticipated low participation rates with e-forms, and the disadvantage of non-response bias. This approach was therefore chosen to maximize the number of participants reached. In addition, non-interventional research ethics committee permission was obtained from [Information removed for blind review] University Ethics Committee (Decision number: E-60116787-020-637605). The research was conducted in accordance with the Declaration of Helsinki ethical principles.

### 2.3. Measures

In this study, Perceived Freedom in Leisure Scale, which is part of the Leisure Diagnostic Battery developed by [24] ([24]) and adapted into Turkish by [96] ([96]), the Leisure Satisfaction Scale developed by [9] ([9]) and adapted into Turkish by [32] ([32]), and the Personal Information Form (PIF) developed by the researcher were used as data collection tools.

#### 2.3.1. Perceived Freedom in Leisure Scale (PFLS)

The PFLS, which was used to collect data in the study, is a 5-point Likert-type scale with 17 items. The Turkish scale has two sub-dimensions: Knowledge/Skill and Excitement/Amusement. A high score on the scale indicates that the perceived freedom is also high ([97]).

Validity and Reliability of Perceived Freedom in Leisure Scale: [24] ([24]) found that the internal consistency coefficient of the unidimensional scale designed to measure perceived competence, perceived control, and perceived intrinsic motivation in leisure was 0.91, and the total mean score was 3.75. In another validity and reliability study conducted by [93] ([93]), the correlations between each item and the total score were found to be greater than 0.45. As a result of the factor analysis conducted by [96] ([96]), who conducted the adaptation study of the scale into Turkish, to determine the construct validity, a 17-item scale was obtained in which two factors explained 47% of the variance.

While the internal consistency coefficient for the entire scale was determined to be 0.90, it was calculated to be 0.80 for the knowledge and skill dimension and 0.80 for the “excitement and fun” dimension ([97]). The Cronbach alpha value as the internal consistency coefficient calculated in this study was 0.90 for the total scale, 0.83 for the knowledge/skill dimension, and 0.78 for the excitement/amusement dimension.

#### 2.3.2. Leisure Satisfaction Scale (LSS)

This is a 5-point scale consisting of 24 questions and 6 sub-dimensions (Psychological, Sociological, Physical, Educational, Relaxation and Esthetic). A high score on the scale also indicates that leisure satisfaction is high ([32]).

Validity and Reliability of Leisure Satisfaction Scale: In the item analysis of the Turkish validity of the scale, it was found that the six-factor structure was confirmed. According to the Pearson product moment coefficient, conducted to test the content validity, significant relationships were found in all subscales (at *p* < 0.001 level, ranging from 0.29 to 0.78). In the reliability analysis conducted by Beard and Ragheb in 1980 for the long form of the scale, it was found that the Cronbach alpha coefficient for each dimension ranged from 0.85 to 0.96. In the short form, the average of this coefficient was 0.93 ([31]). The Cronbach alpha value as the internal consistency coefficient calculated in this study was found to be 0.91 for the total scale. In its subdimensions, it was found to vary between 0.67 and 0.80.

#### 2.3.3. Personal Information Form (PIF)

This questionnaire was designed by researchers to obtain information about the study’s independent variables. In the form, there are questions about the participants’ gender, age, how they participate in leisure activities, and what settings they prefer to participate in frequently.

### 2.4. Statistical Analysis

Data were analyzed using various statistical techniques, including frequency analysis, arithmetic mean, standard deviation, Pearson correlation analysis, linear regression analysis, three-factor ANOVA, one-way MANOVA, and factorial (3 × 2) MANOVA. Pearson correlation analysis was used to determine the relationships between the subdimensions and total scores of perceived freedom in Leisure Scale and Leisure Satisfaction Scale. Linear regression analysis assessed whether PFLS could predict LSS. Differences in total LSS based on the type of leisure participation and the setting of participation were evaluated using a three- factor ANOVA. One-way MANOVA was utilized to analyze differences in the subdimensions of PFLS and LSS by gender. In addition, a factorial (3 × 2) MANOVA was used to assess differences in the subdimensions of PFLS based on participation type and field. Whether the data met the prerequisites for parametric tests was determined by examining the values of skewness and kurtosis (normal distribution status of the data) and the results of the Levene test (equality of variances) ([84]). It was found that the skewness values for the total scores and subdimensions of the measurement tools were between −0.463 and −0.670, and the kurtosis values were between 0.151 and 0.615. The scatterplot was used in correlation and regression analyses to determine whether the relationship between the variables was linear. In the MANOVA analysis, Box’s test was used to test the assumption that the covariance matrices of the dependent variables were equal across groups. In the same analysis, Wilk’s Lambda test was used to test the difference in the means of the dependent variables according to the groups in the independent variables. Cronbach Alpha internal consistency coefficients were calculated to determine the reliability of the scales used in the study. The type 1 error was accepted as 5%.

## 3. Results

The relationship between perceived freedom in leisure and leisure satisfaction was examined using Pearson correlation analysis. The detailed correlation coefficients between the sub-dimensions of the PFLS and the LSS are presented in Table 1.

As presented in Table 1, moderate and statistically significant positive correlations were found between the total and sub-dimension scores of the PFLS and the LSS.

The strongest association was observed between the total PFLS and total LSS scores (r = 0.65; *p* < 0.01), while the weakest correlation is between the knowledge and skill sub-dimension of the PFLS and the “physiological” sub-dimension of the LSS (r = 0.42; *p* < 0.01).

These findings indicate that as students’ perceived freedom in leisure increases, their overall satisfaction with leisure experiences also tends to increase.

To determine whether perceived freedom in leisure predicts leisure satisfaction, a linear regression analysis was performed. The results of this analysis are presented in Table 2.

As shown in Table 2, the regression analysis indicates that perceived freedom in leisure is a significant predictor of leisure satisfaction. (R = 0.651; R^2^ = 0.424; F (1, 3190) = 2350.973; *p* < 0.001). According to the R^2^ value, perceived freedom in leisure accounts for approximately 42% of the total variance in leisure satisfaction scores among students. The regression equation derived from the model is as follows: leisure satisfaction = 1.373 + (0.648 × perceived freedom in leisure).

To explore whether perceived leisure freedom and leisure satisfaction differed by gender, a one-way MANOVA was conducted. The detailed results of this analysis are presented in Table 3.

As shown in Table 3, significant gender-based differences were found in several sub-dimensions of both the PFLS and the LSS (Wilks’ Lambda = 0.972; F = 11.347; *p* ≤ 0.001).

According to the follow-up ANOVA results, no significant differences were observed in the knowledge/skill sub-dimension of the PFLS (F = 0.034, *p* = 0.853) or the physical sub-dimension of the LSS (F = 0.003, *p* = 0.953).

However, female participants reported significantly higher scores than male participants in the excitement/amusement sub-dimension of the PFLS (*µ*_female_ = 4.03; *µ*_male_ = 3.92), as well as in the following LSS sub-dimensions: psychological (*µ*_female_ = 3. 96; *µ*_male_ = 3.85), educational (*µ*_female_ = 4.08; *µ*_male_ = 3.98), social (*µ*_female_ = 3.99; *µ*_male_ = 3.93), relaxation (*µ*_female_ = 4.13; *µ*_male_ = 4.03), and esthetic (*µ*_female_ = 4.39; *µ*_male_ =3.83).

To examine how different leisure participation styles and settings influence overall leisure satisfaction, a three-factor ANOVA was performed. The results of this analysis are presented in Table 4.

As shown in Table 4, the interaction of active/passive × individual/group participation in leisure activities was found to be significant (F = 14.116; *p* = < 0.001). Accordingly, when the line graph was examined, it was seen that the LSS of the active participants in leisure was higher than the passive participants, whether individually or in a group, but the scores of those who participated in passive and individual activities were the lowest (*µ*_active/individual_ = 4.01; *µ*_active/group_ = 3.95; *µ*_passive/individual_ = 3.71; *µ*_passive/group_ = 3.83).

The main effects of active/passive and indoor/outdoor leisure participation on LSS were also found to be significant (*µ*_active/passive_ = 84.854, *p* = < 0.001; *µ*_outdoor/indoor_ = 4.004, *p* = 0.045). Compared to passive participants, active participants reported higher levels of leisure satisfaction. In addition, participants in outdoor settings demonstrated slightly higher LSS scores than those in indoor settings (*µ*_active_ = 3.98; *µ*_passive_ = 3.77; *µ*_outdoor_ = 3.90; *µ*_indoor_ = 3.85).

No significant differences were found in leisure satisfaction scores based on the main effects or interactions of active/passive × indoor/outdoor setting, individual/group × indoor/outdoor setting, active/passive × individual/group × indoor/outdoor setting, or individual/group participation alone.

To examine differences in perceived freedom in leisure across participation styles and settings, a 3 × 2 MANOVA was conducted. The ANOVA results for each dependent variable are presented in Table 5.

As shown in Table 5, the 3 × 2 MANOVA revealed significant main effects of active/passive participation (λ = 0.969, F = 51.214, *p* < 0.001), individual/group participation (λ = 0.996, F = 5.631, *p* = 0.004), and indoor/outdoor setting (λ = 0.997, F = 5.408, *p* = 0.005) on the sub-dimensions of the PFLS.

In terms of interaction effects, a significant three-way interaction was observed among active/passive, individual/group, and indoor/outdoor variables (λ = 0.998, F = 3.092, *p* = 0.046). In addition, the two-way interaction between individual/group and indoor/outdoor participation was also significant (λ = 0.998, F = 3.088, *p* = 0.046). However, the interaction between active/passive and individual/group participation (λ = 0.999, F = 2.162, *p* = 0.115), as well as the interaction between active/passive and indoor/outdoor settings (λ = 0.998, F = 3.088, *p* = 0.046), did not reach statistical significance.

Follow-up ANOVA tests were conducted to determine which independent variables contributed to the multivariate significance of the dependent variables.

Table 5 shows that there are no significant differences in the “Knowledge/Skill” sub-dimension based on the three-way interaction of active/passive × individual/group × indoor/outdoor settings (F = 1.199, *p* = 0.274). However, this interaction produced a significant difference in the excitement/amusement sub-dimension (F = 5.036, *p* = 0.025). Confidence interval analysis revealed that participants who engaged in active, group-based leisure activities in outdoor settings scored significantly higher (*µ* = 4.03) than those who participated passively, individually, and in indoor settings (*µ* = 3.77).

Although the interaction between active/passive × individual/group participation was not significant for excitement/amusement (F = 1.531, *p* = 0.216), it was significant for knowledge/skill (F = 4.030, *p* = 0.045). Line graph analysis showed that active participants consistently scored higher than passive participants, regardless of whether they participated individually or in groups. Among the four combinations, the highest scores were reported by those participating actively in groups (*µ* = 4.03), and the lowest by those participating passively and individually (*µ* = 3.71). The overall mean scores were as follows: active/individual = 4.00, active/group = 4.03, passive/individual = 3.71, and passive/group = 3.84. For the knowledge/skill sub-dimension, no significant interaction was found between active/passive × indoor/outdoor settings (F = 0.022, *p* = 0.883) or between individual/group × indoor/outdoor settings (F = 0.846, *p* = 0.358).

In contrast, for the excitement/amusement sub-dimension, the active/passive × indoor/outdoor interaction was not significant (F = 0.446, *p* = 0.504), while the individual/group × indoor/outdoor interaction showed a significant effect (F = 4.639, *p* = 0.031). According to the line graph, the highest scores were observed among those who participated in outdoor activities in groups (*µ* = 3.99), whereas the lowest scores were re-ported by those who participated individually in outdoor activities (*µ* = 3.86). In both settings, participants involved in group-based activities scored higher than those who participated individually (*µ*_group/indoor_ = 3.89; *µ*_individual/indoor_ = 3.87).

Regarding the main effects of the independent variables on the dependent variables, no significant difference was observed in the knowledge/skill sub-dimension based on the indoor/outdoor setting (F = 0.006, *p* = 0.937). However, significant differences were found for both sub-dimensions according to the other independent variables.

In the knowledge/skill sub-dimension, participants who were actively involved in leisure activities scored higher on average than those who participated passively (*µ*_active_ = 4.00; *µ*_passive_= 3.75). Similarly, group participants reported higher scores than individual participants (*µ*_individual_ = 3.83; *µ*_group_ = 3.93).

A similar pattern was observed in the excitement/amusement sub-dimension, where active participants had higher mean scores than passive ones (*µ*_active_ = 3.99; *µ*_passive_ = 3.79). Furthermore, group participants outperformed individual participants (*µ*_individual_ = 3.84; *µ*_group_ = 3.93), and participants in outdoor settings scored higher than those in indoor settings.

## 4. Discussion

This descriptive and correlational study aimed to determine the predictive relationship between levels of perceived freedom in Leisure (PFL) and Leisure Satisfaction (LS) among students in the field of sports sciences. The study also compared these two variables based on factors related to leisure participation.

### 4.1. Predictive Relationship Between Perceived Freedom in Leisure and Leisure Satisfaction

The findings revealed a positive relationship between PFL and LS, indicating that PFL is a significant predictor of LS. Previous research supports this conclusion, reporting consistent evidence of a relationship between the two variables ([79]; [54]; [95]; [2]; [77]). In addition, it has been observed that a lack of freedom in leisure leads to lower satisfaction ([65]). [15] ([15]) also found a negative relationship between leisure constraints opposite to perceived freedom in leisure and leisure satisfaction. Furthermore, young people who experience sufficient freedom in their leisure activities tend to enjoy all aspects of life more ([48]) and have higher expectations for participation, commitment, and value in their leisure activities ([66]). This situation is believed to contribute to greater leisure satisfaction. This finding can also be interpreted within the framework of self-determination theory, which posits that autonomy and perceived control enhance intrinsic motivation and satisfaction. From a psychological perspective, the ability of individuals to choose activities based on their own preferences fulfills their needs for intrinsic motivation and autonomy ([75]). As predicted by self-determination theory ([21]), this process leads to higher levels of satisfaction and well-being. Therefore, the predictive role of PFL on LS is consistent with this theoretical framework. In the Turkish context, where university students may face socio-cultural constraints such as family structures and peer relationships that limit their leisure options ([33]), perceived freedom becomes particularly critical for enhancing leisure satisfaction. Accordingly, universities and policymakers should design leisure programs that provide students with a variety of activity options tailored to different interests and needs, thereby enhancing their overall well-being and leisure satisfaction through greater freedom of choice.

### 4.2. Gender-Based Differences in Perceived Freedom in Leisure

The study found that both female and male students had comparable knowledge and skill levels in PFL concerning gender differences; however, female students reported feeling more excitement and enjoyment than male students. Overall, female students tended to be more satisfied with their leisure activities than male students. This aligns with [92]’s ([92]) framework, which suggests that while some leisure activities meet similar needs across genders, others fulfill distinct psychological needs. Specifically, males tend to gravitate toward activities that provide safety and tangible rewards, whereas females prefer leisure experiences that emphasize enjoyment, collaboration, and shared decision-making. This finding supports Kelly’s (1978, as cited in [91]) proposition that intrinsic motivations enhance the perception of freedom; however, these intrinsic motives may differ by gender. As a result, male students’ leisure engagement appears to be more influenced by external or environmental factors, potentially leading to lower levels of excitement and enjoyment compared to female students.

There are inconsistent findings in the literature regarding gender-based differences in perceived freedom in leisure (PFL). While some studies indicate that women report higher levels of PFL ([95]; [41]), others have found no significant differences ([97]; [62]). [8] ([8]) and [86] ([86]) noted that there may be gender differences in desires, goals, and preferences related to leisure activities, and that individuals of different genders may experience various aspects of leisure, including the perception of freedom, in different social contexts. Similarly, [12] ([12]) emphasize that as individuals age, responsibilities and leisure opportunities associated with gender roles change, leading to variations in leisure experiences.

An important cultural dimension emerges when interpreting our findings in the Turkish context. Previous research highlights that women worldwide, and particularly in patriarchal societies like Turkey, face greater intrapersonal and interpersonal leisure constraints ([50]; [91]), including family obligations, social approval, safety concerns, and transportation issues. “Care ethics,” reflecting expectations of women’s caregiving roles, is one of the most prominent factors limiting women’s leisure both in Turkey and around the globe ([50]). However, in this study, female university students do not yet carry responsibilities such as childcare or household management, allowing them to experience greater autonomy during their university years. Leaving their family homes to study in different cities may provide opportunities to escape some traditional constraints, fostering higher perceptions of freedom. This could help explain why women in this study reported more excitement and enjoyment than men.

On the other hand, patriarchal norms remain deeply rooted, particularly among lower socio-economic groups, and both women and men tend to internalize these gender roles, which are then reflected in their leisure behaviors. The literature frequently emphasizes that women are disadvantaged compared to men in terms of access to leisure opportunities, although only a limited number of studies are cited here ([40]; [22]; [35]; [34]). In this context, future longitudinal research (e.g., comparing PFL measurements before and after university) could help determine whether women’s higher perceptions of freedom are associated with the transition to university life and living in different cities.

The findings of this study are consistent with the growing body of international research on women’s empowerment and resistance through leisure ([35]). In the Turkish context, however, men may experience greater societal pressure to prioritize responsibilities such as work or academic performance, which can reduce their perceived autonomy in leisure time. [50] ([50]) noted that, under traditional Turkish law, men were regarded as the primary authority within the household and were legally obliged to provide care for their children and spouse. Although such legal regulations have since been revised, the patriarchal structure continues to influence social roles for both women and men. As [40] ([40]) emphasized, participation in leisure is shaped not only by biological sex but also by the cultural and situational contexts of gender. Therefore, developing a culturally sensitive perspective is essential when interpreting gender differences in perceived freedom in leisure (PFL). Accordingly, campus recreation programs at universities should offer diverse and safe activity options that foster intrinsic enjoyment and collaboration for women, while also reducing external pressures for men. Such gender-sensitive policies can enhance equitable access and meaningful participation for all students.

### 4.3. Gender-Based Differences in Leisure Satisfaction

Research proved that female and male students reported similar satisfaction in the physiological sub-dimension, i.e., the health gain from their leisure activities. Overall, however, female students were more satisfied with their leisure activities than male students. There is some literature with varied findings on this topic. In studies of university students by [61] ([61]), of users of park recreation areas by [95] ([95]), of gay, lesbian, bisexual, and transgender by [69] ([69]), and of couples by [79] ([79]), gender was found to have no impact on outcomes. However, [63] ([63]) concluded that male students experienced psychological impacts more positively on leisure satisfaction and felt more comfortable than female students, based on their research among university students. The same study did not identify the other sub-dimensions as differing by gender ([63]). [31] ([31]) study in Turkey found that the female students were more satisfied with leisure in the relaxation sub-dimension and male students were more satisfied with the physiological sub-dimension. There was no gender difference in the remaining sub-dimensions. On the contrary, a study by [67] ([67]) of South African university students found a gender difference that was significant in favor of female students. In line with their study, while these findings of the study may be the same in different samples, they may be explained by differences in gender equity in different societies. In societies with gender equality, male and female students have the same opportunities to use leisure resources ([67]). However, this is the opposite of most evidence demonstrating that girls and women are typically disadvantaged since they are hampered by household chores and structural gaps in leisure opportunity access ([28]; [40]; [34]). This is a continuous and serious issue that should not be overlooked.

Within the Turkish cultural context, traditional values and geographic diversity are important factors in how leisure experiences are formulated. Although leisure might be broadly described as a space for personal development, within contexts that are dominated by traditional norms, it tends to represent a time devoted to family and kinship responsibilities rather than as time for personal freedom ([33]). Such traditional lifestyles lived in rural areas and spontaneous suburban dwellings are very different from urban lifestyle. Even their outdoor activities vary. There are more nature-oriented and community-oriented activities in rural settings, while in cities, there are more sports, entertainment, and technology-oriented activities to choose from ([6]; [25]). Students from different regions who attend universities in urban centers are often exposed to new opportunities for leisure. Female students, who before might have been more exposed to conventional pressures and patriarchal limitations while residing with their families ([50]), are now confronted with a broader range of leisure activities that respond to diverse needs and interests. This exposure can increase their perceived leisure freedom and hence their level of satisfaction. Supporting this, [50] ([50]) highlighted the fact that as a part of the general social transformations of Turkey, opportunities for women and concepts they attribute to leisure, particularly sports, have shifted, with women of diverse backgrounds in urban areas participating more in sports and fitness.

These cultural and theoretical explanations may account for the reason why female students of sports science in Turkey scored higher on leisure satisfaction compared to their male counterparts. The findings also have important practical implications for university recreation programs and policymakers. Designing inclusive activities that address physiological, relaxation, and social dimensions, as well as providing culturally sensitive, safe, and accessible facilities alongside flexible scheduling opportunities, can enhance student well-being and promote greater equity in leisure experiences.

In addition to the explanations above, these findings highlight the consistency between the two main variables examined in the study and the internal coherence of the research. The gender difference observed in LS was also reflected in the “Excitement and Enjoyment” subdimension of PFL, which is structurally and conceptually more closely related to LS. In other words, female students not only derived greater satisfaction from their leisure time but also perceived these experiences as more exciting and enjoyable. Since previous analyses demonstrated PFL as a significant predictor of LS, it is theoretically consistent and expected that the difference in satisfaction is supported by a similar difference in the key predictor variable.

### 4.4. Differences in Perceived Freedom and Leisure Satisfaction by Participation Type and Setting

Research findings indicate that actively participating in leisure activities, engaging in social environments rather than solitary ones, and taking part in outdoor activities positively contribute to both LS and PFL. Mannell (1984) also noted that activities that provide personal satisfaction in leisure are typically those in which individuals actively participate (Mannell 1984 as cited in [91]). [11] ([11]) found that individuals engaged in physical activity, an active form of exercise, reported higher levels of PFL. Similarly, an experimental study revealed that participating in eight weeks of sportive recreational activities had a positive impact on PFL, reinforcing the notion of a cause-and-effect relationship ([26]). In addition, [94] ([94]) conducted a study with university students and discovered an asymmetrical relationship between PFL and passive activities. They also noted that restricted and passive activities can impede the holistic intellectual, psychological, and social development of young people. University students tend to favor indoor, individual, and less physically active pursuits, such as watching TV or videos and shopping. This preference can lead to a lack of engagement in social environments, negatively impacting their social development ([94]). As a result, an imbalanced relationship may develop between the perceived freedom to choose activities and the tendency to engage in passive ones. This finding should be interpreted within the specific socio-cultural context of Taiwan, since each society has unique factors shaping leisure behaviors. Therefore, direct generalization of these results to the Turkish context may be limited. Nevertheless, the observation that passive and constrained activities can negatively influence social development and perceptions of freedom across different cultures suggests that there may be universal aspects in this relationship. According to [65] ([65]), the type of leisure activity influences both the amount of time devoted to leisure pursuits and intrinsic factors such as satisfaction, motivation, and the sense of freedom associated with leisure. Research indicates that passive activities, such as watching television, using harmful substances, and engaging in aimless conversations, are linked to an increase in criminal behavior among young people ([27]). In contrast, active activities (such as participating in sports or performing in the arts) are more effective in fostering social competence, self-esteem, and acceptance within larger social groups. Passive activities do not promote psychological and social adaptive behaviors ([68]). The literature also shows a positive correlation between increased perceived freedom in leisure and higher active participation in exercise or sports. Conversely, individuals who cannot engage in physical activities an important aspect of active leisure often experience a diminished sense of freedom ([70]; [71]; [38]; [59]). These findings are also consistent with the theoretical framework proposed by [48] ([48]), who categorized leisure activities into two types: relaxed leisure, involving passive activities, and serious leisure, requiring effort and including active activities that provide developmental benefits. They emphasized that serious leisure more strongly supports individuals’ freedom of choice, intrinsic motivation, and personal growth. This framework conceptually explains the positive impact of active participation on PFL and LS observed in this study.

On the other hand, group activities involving social interaction have a positive impact on the perception of leisure freedom ([94]). Group activities, such as team sports, foster social satisfaction through various processes, including forming friendships, expanding social networks, developing group identity, and creating a sense of belonging ([71]). [3] ([3]) found that engaging in activities with others, rather than experiencing them alone, enhances leisure satisfaction. Similarly, [78] ([78]) highlighted that social leisure fosters social relationships, emphasizing that young people have a need to establish positive peer connections and tend to devote more time to social activities. This phenomenon can also be explained within the framework of Self-Determination Theory. In this context, it can be inferred that participating in leisure activities as a group allows sports sciences students to feel socially supported through their own autonomous choices. Through social interaction and exposure to diverse experiences, they may have developed a sense of competence and fostered a feeling of belonging by communicating with those around them ([21]). These dynamics help explain why participants in this study reported higher levels of PFL and LS when engaging in group rather than individual activities.

These findings are also meaningful when evaluated within the Turkish context. In Turkish society, where family and kinship relations traditionally play a central role, young people’s participation in leisure activities is often shaped by social contexts ([33]). In lower socio-economic groups dominated by traditional structures, youth may face social pressures to prioritize academic achievement or family responsibilities over leisure pursuits. Such pressures can lead to greater engagement in passive activities while limiting active participation. On the other hand, students who begin university life and move to different cities away from their families encounter a wider range of leisure alternatives, which can enhance their participation in both outdoor and group-based activities. In this regard, the psychological relaxation, sense of belonging, and social support provided by outdoor and group activities elevate students’ levels of perceived freedom and satisfaction. Particularly in countries like Turkey, where urbanization is rapidly intensifying, ensuring opportunities for interaction with natural environments should be considered an important policy priority for enhancing youth psychological well-being. Therefore, university recreation programs and policymakers should develop inclusive strategies that respond to students’ diverse interests and needs by increasing the diversity of safe and accessible outdoor events and group-based recreational activities. Such initiatives would not only support individual well-being but also foster social cohesion and culturally sustainable recreation policies.

## 5. Conclusions

This study demonstrates that perceived freedom in leisure (PFL) was a powerful predictor of leisure satisfaction (LS) among sports sciences students and accounted for nearly half of the variance. Female students recorded higher levels of both PFL and LS, while active, group-based, and outdoor activities were highly correlated with greater freedom and satisfaction perceptions. These findings reinforce self-determination theory, highlighting the central role of autonomy and intrinsic motivation in enhancing leisure experiences. They are also consistent with [48]’s ([48]) distinction between serious and relaxed leisure, showing that effortful and socially engaging activities are more effective than passive pursuits in fostering freedom and satisfaction.

The originality of this research is in moving beyond the well-established psychosocial benefits of LS and PFL to examine systematically the role of participation style (individual/group, active/passive) and setting (outdoor/indoor) in shaping them. In particular, the establishment of outdoor activities as providing greater freedom and satisfaction advances a relatively underexplored section of the literature.

Set in Turkey’s socio-cultural context, which is characterized by tight-knit family structures, gender roles, and community-based relations, the findings emphasize the importance of culturally adaptable leisure activities that promote autonomy, pleasure, and relatedness. At the same time, identifying those conditions that systematically enhance PFL and LS offers conceptual gains with wider applicability, guiding future research and the development of recreation programs and youth policy in intricate cultural contexts.

## 6. Limitations and Recommendations

Although the study has some strengths such as the number of participants, the measurement tools used, etc., it also has some limitations. Explanation of these limitations and some suggestions for future studies may provide convenience for researchers who will want to carry out studies in this field. One of the limitations is that the participants were only undergraduate students from the field of sport sciences. It may be useful to examine leisure activities in lower and upper age groups in a cause-and-effect relationship with LS and PFL for a more comprehensive description of the subject. The second limitation is that the study was based on a survey; leisure organizations were not arranged, and no intervention programs were implemented. Future research could conduct pre- and post-evaluations by encouraging university students to participate in voluntary, budgeted, or free activities. The outcomes of such studies may draw the attention of university sports and culture departments and highlight students’ perceptions of freedom and satisfaction with campus facilities. Another limitation is that there is no data on another variable that mediates between the LS and PFL variables that we examined in a cause–effect relationship. The effect of many psycho-social variables that may mediate between these two variables can be considered. Furthermore, this study was conducted exclusively with university students in Turkey, and the country’s unique socio-cultural structure, family relationships, gender roles, and dynamics of university life may have influenced students’ perceptions and behaviors related to leisure. Therefore, the findings have limited direct generalizability to different cultural and social contexts. This underscores the importance of conducting comparative studies in diverse cultural settings. Future research should include multi-site studies involving university students from various countries, take cultural variables into account, and employ mixed-method designs supported by qualitative approaches.

### 6.1. Recommendations for Practice and Policy

#### 6.1.1. Curriculum and Program Development

Curricula can be enriched with both theoretical and practical courses (e.g., recreational sports, outdoor activities, arts and sports, leadership and organization training) that meet students’ psychological needs such as excitement, enjoyment, social interaction, and esthetic experiences during their leisure time.

Findings indicate that participants perceived greater freedom and satisfaction when engaging in outdoor leisure activities rather than indoor activities. Accordingly, recreational opportunities on campus should not be limited to indoor spaces; instead, outdoor recreational facilities should also be developed. Investments can be made in open-air sports facilities, trails, adventure parks, and socially interactive spaces, and their safe use can be promoted by integrating them into curricula.

Since findings revealed gender differences in PFL and LS, targeted programs that encourage inclusive leisure experiences for all students are essential. Club activities, special incentive programs, and safe participation environments should be created to meet diverse leisure needs, allowing students to freely design their own leisure experiences.

Furthermore, an elective “Leisure Literacy” course could be introduced across all faculties, not only within sports sciences. The purpose of this course would be to help students identify their sources of freedom and satisfaction, develop skills for planning activities, and improve their access to community-based recreational opportunities.

A flexible and modular pool of activities that fosters knowledge and skill development could also be created. Beyond the traditional activities they may have previously experienced, students could be encouraged to participate in diverse areas such as mountaineering, photography, street games, digital gaming leagues, ceramics, dance, and volunteering projects.

#### 6.1.2. Program Administrators

Findings clearly demonstrate that active participation significantly enhances leisure satisfaction compared to passive participation. Program administrators can design initiatives that transform individuals from spectators into active participants (e.g., amateur leagues, public races, workshops, skill development camps).

Content should be organized to provide students with “hands-on” experiential learning opportunities. Initiatives led by students that foster autonomy can be encouraged. Instead of relying solely on administration-organized activities, systems allowing students to design and manage their own events (e.g., small grant support, mentorship) could be implemented. Creating their own activities is likely to provide the highest perceived leisure freedom.

Similarly to career or life coaching, a “Leisure Coaching” service could be developed to help individuals build a more satisfying leisure life. Coaches could assess students based on the sub-dimensions identified in this study and guide them toward activities that meet their psychological, social, physiological, or esthetic needs.

#### 6.1.3. Policymakers

In Turkey’s national development plans, which are designed to promote progress in areas such as economy, health, education, transportation, social security, and justice, leisure should be addressed strategically as a critical component of public health.

In this context, leisure should be planned for all segments of society, and specifically for young people, with an approach that ensures gender equality in opportunities. Policies should include diversifying outdoor activity options, ensuring safety in these spaces, and developing programs that encourage social participation. These strategies aim to enhance leisure well-being while promoting inclusive and sustainable recreation policies at the societal level.

#### 6.1.4. Scientific and Academic Implications

Future research could examine how Turkey’s rich geographical and cultural characteristics are reflected in regional leisure policies, opportunities, practices, and experiences.

Qualitative research methods can explore why women score higher in certain sub-dimensions or through which mechanisms outdoor activities enhance satisfaction.

Furthermore, longitudinal studies that follow students during the transition from high school to university, moving from living with family to independent living in a different city, could provide clearer insight into potential changes in perceived leisure freedom and satisfaction among female students in Turkey.

Intervention programs that enhance perceived freedom (e.g., providing choice, supporting autonomy) could be developed and tested for their effects on satisfaction. Measurements can also be repeated with special needs and disadvantaged groups to compare results and design campus recreation activities tailored to their needs.

#### 6.1.5. Digital Transformation and Technology Integration

Virtual and augmented reality (VR/AR) applications offer significant opportunities to transform the traditional spectator experience. In the long term, recreational or outdoor sports activities enriched with VR/AR technologies could provide users with immersive, interactive experiences where they feel virtually “on the field.” This approach may create new opportunities for individuals with limited access and serve as an innovative way to experience traditional sports. It also contributes to integrating leisure experiences into the lives of new generations in an ever-changing world.

Personalized recreation apps designed as smartphone-based “leisure recommendation engines” could match users’ interests, skill levels, and current moods. For instance, the app could analyze a user’s PFL and LS profile and recommend activities likely to provide the highest satisfaction (e.g., “Today your mood is better suited for a group outdoor activity”).

Furthermore, “Sports and Recreation Networking Platforms” could be developed to connect individuals with similar leisure interests. This may be particularly valuable for those who have moved to a new city or wish to expand their social circles.

These recommendations are tailored to Turkey’s cultural context and should be adapted cautiously when applied to other societies.

## Figures and Tables

**Table 1 behavsci-15-01273-t001:** Pearson correlation analysis of perceived freedom in leisure and leisure satisfaction among sports science students.

n = 3192	Psychological	Educational	Sociological	Relaxation	Physical	Esthetic	LSS
Knowledge/Skill	0.48 **	0.49 **	0.50 **	0.45 **	0.42 **	0.43 **	0.59 **
Excitement/Amusement	0.46 **	0.55 **	0.55 **	0.52 **	0.44 **	0.45 **	0.64 **
PFLS	0.49 **	0.55 **	0.56 **	0.51 **	0.46 **	0.47 **	0.65 **

** *p* < 0.01.

**Table 2 behavsci-15-01273-t002:** Results of linear regression analysis predicting leisure satisfaction based on perceived freedom in leisure.

Variable	B	Standard Error B	β	t	*p*
Constant	1.373	0.053	0.651	25.702	*p* < 0.001
PFLS	0.648	0.013		48.487	*p* < 0.001
R = 0.651; R^2^ = 0.424; Adjusted R^2^ = 0.424					
F(1, 3190) = 2350.973; *p* ≤ 0.000					

**Table 3 behavsci-15-01273-t003:** Results of one-way MANOVA comparing perceived leisure freedom and leisure satisfaction among sports science students by gender.

Source of Variance	Dependent Variable	Sum of Squares	df	Mean Square	F	*p*	ηp2
Gender	Knowledge/Skill	0.010	1	0.010	0.034	0.853	0.000
	Excitement/Amusement	7.727	1	7.727	21.745	*p* < 0.001	0.007
	Psychological	8.586	1	8.586	20.304	*p* < 0.001	0.006
	Educational	7.175	1	7.175	15.208	*p* < 0.001	0.005
	Sociological	2.533	1	2.533	5.555	0.018	0.002
	Relaxation	7.165	1	7.165	14.853	*p* < 0.001	0.005
	Physical	0.002	1	0.002	0.003	0.953	0.000
	Esthetic	4.995	1	4.995	9.041	0.003	0.003
Eror	Knowledge/Skill	975.402	3190	0.306			
	Excitement/Amusement	1133.592	3190	0.355			
	Psychological	1348.995	3190	0.423			
	Educational	1504.970	3190	0.472			
	Sociological	1454.636	3190	0.456			
	Relaxation	1538.715	3190	0.482			
	Physical	1552.847	3190	0.487			
	Esthetic	1762.446	3190	0.552			
Total	Knowledge/Skill	1762.446	3192				
	Excitement/Amusement	51,402.563	3192				
	Psychological	49,895.813	3192				
	Educational	53,099.438	3192				
	Sociological	51,381.688	3192				
	Relaxation	54,539.625	3192				
	Physical	48,289.000	3192				
	Esthetic	49,509.563	3192				

**Table 4 behavsci-15-01273-t004:** Results of a three-factor ANOVA examining differences in total leisure satisfaction scores among sports sciences students based on leisure participation styles and participation setting.

Source of Variance	Sum of Squares	df	Mean Square	F	*p*	ηp2
1. Active/Passive	23.973	1	23.973	84.854	*p* < 0.001	0.026
2. Individual/Group	0.572	1	0.572	2.025	0.155	0.001
3. Indoor/Outdoor	1.131	1	1.131	4.004	0.045	0.001
1 × 2	3.988	1	3.988	14.116	*p* < 0.001	0.004
1 × 3	0.386	1	0.386	1.368	0.242	0.000
2 × 3	0.676	1	0.676	2.393	0.122	0.001
1 × 2 × 3	0.011	1	0.011	0.040	0.842	0.000
Error	899.543	3184	0.283			
Total	50,492.909	3192				

**Table 5 behavsci-15-01273-t005:** ANOVA results of the 3 × 2 MANOVA test examining differences in the sub-dimensions of perceived freedom in leisure based on participation type and setting among physical education students.

Source of Variance	Dependent Variable	Sum of Squares	df	Mean Square	F	*p*	ηp2
1.Active/Passive	Knowledge/Skill	30.082	1	30.082	102.075	*p* < 0.001	0.031
	Excitement/Amusement	19.849	1	19.849	56.846	*p* < 0.001	0.018
2.Individual/Group	Knowledge/Skill	3.265	1	3.265	11.079	0.001	0.003
	Excitement/Amusement	2.893	1	2.893	8.284	0.004	0.003
3.Indoor/Outdoor	Knowledge/Skill	0.002	1	0.002	0.006	0.937	0.000
	Excitement/Amusement	1.365	1	1.365	3.909	0.048	0.001
1 × 2	Knowledge/Skill	1.188	1	1.188	4.030	0.045	0.001
	Excitement/Amusement	0.535	1	0.535	1.531	0.216	0.000
1 × 3	Knowledge/Skill	0.006	1	0.006	0.022	0.883	0.000
	Excitement/Amusement	0.156	1	0.156	0.446	0.504	0.000
2 × 3	Knowledge/Skill	0.249	1	0.249	0.846	0.358	0.000
	Excitement/Amusement	1.620	1	1.620	4.639	0.031	0.001
1 × 2 × 3	Knowledge/Skill	0.353	1	0.353	1.199	0.274	0.000
	Excitement/Amusement	1.759	1	1.759	5.036	0.025	0.002
Error	Knowledge/Skill	938.346	3184	0.295			
	Excitement/Amusement	1111.770	3184	0.349			
Total	Knowledge/Skill	51,058.469	3192				
	Excitement/Amusement	51,402.563	3192				

## Data Availability

The datasets generated and/or analyzed during the current study are available from the corresponding author on reasonable request (otepekoylu@pau.edu.tr).

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
