# Peer review of "Exploring the Interplay of Leisure Freedom, Satisfaction, and Participation Styles Among Turkish Sports Sciences Students"

_behavsci, 2025, doi:10.3390/bs15091273_

Round 1

Reviewer 1 Report

Comments and Suggestions for Authors

Dear Authors,

This study explores an interesting and relevant topic, with solid effort in design and analysis. Separating the Introduction and Theoretical Background, deepening the Discussion, and strengthening the Conclusion will make the work clearer and more impactful.

1. Introduction & Theoretical Background (currently integrated in one section)

- The introduction and theoretical background are integrated into one section without distinction, making it difficult for readers to understand the logical flow, and the introduction and theoretical background should be separated.

- The introduction needs to summarize the purpose and necessity of the research, the organization and differentiation of previous research, and an explanation of what problems have not been solved in the previous research

- The theoretical background clearly presents the main variables and academic definitions of concepts covered in the research, and explains the definitions adopted by the researcher by comparing and organizing the views of various scholars to explain the definitions adopted by the researcher and the reasons for them.

2. Methods

- Overall clear and systematic

3. Results

- Statistical analysis is substantial

4. Discussion

- The current discussion is limited to a 'same/different' level comparison with existing research. Interpretation of results should be linked to causal and contextual explanations. There is no policy or practical implication of the research results.

Details:

- When comparing each result with the existing literature, it is necessary to explain why such differences or agreements occurred

- In addition to theoretical implications, it is necessary to present specific practical application methods such as planning and policy formulation for university sports education and leisure programs

- The content of the discussion should be sub-sectioned by topic to clarify the logical flow.

5. Conclusion

- It must be revised, and if it is not corrected, the value of the research will be significantly reduced, and the conclusion currently written is no better than the abstract, and the conclusion must include a significant amount of content related to practical implications.

Details:

(1) The possibility of using the research results is not presented.

There are no specific guidelines on how universities, sports-related institutions, policymakers, and program operators can use this research. (e.g., Curriculum design, outdoor program planning, and strategies tailored to gender and activity type)

(2) Lack of reflection of cultural and regional characteristics

As the study targeted Turkish university students, it is necessary to present social contexts or limitations that can be applied in the country and culture, and there is no mention of generalizability and limitations to other countries or groups.

(3)Research contribution is unclear

The conclusion does not reveal what is new compared to the previous research.

Reviewer 2 Report

Comments and Suggestions for Authors

Thank you for the opportunity to read and review this manuscript. You have produced a very clear and well-written manuscript, but I think you can make some improvements to how the paper is introduced in order to make a better case for the originality and benefit of the paper. Although this may seem like a minor revision, in its current form the manuscript does not adequately highlight a research gap and/or show how it is addressing that, so I would need this to be sufficiently addressed prior to publication. I also feel that some of the conclusions drawn in the discussion don't do justice to the large body of sport/leisure research that could be incorporated to better make some of your points, particularly around the systemic barriers faced by specific populations (e.g. women and girls). Please see specific feedback below. All comments are intended as constructive, and I hope they are received as such:

Introduction

Page 3 Line 112-114: You have noted the 6 dimensions of leisure satisfaction (Beard & Ragheb, 1980), but then not really expanded on these or shown how they are relevant to your research. Please explore these concepts further as I feel they have more to contribute in better setting up your manuscript.

Page 4 Line 151-: Your introduction ends quite abruptly, and there is insufficient rationale as to why your study is necessary (e.g. what gap does it fill) and why you chose to sample sports science students. I would like to see you explicitly state what the research gap is and how your study contributes to it. Were sport science students a convenience sample or was this demographic targeted? Please also clearly list your research questions and/or aims/objectives.

Methods

Page 5 Line 191-195: This is a repetition from the previous page, please remove one of these passages.

Results

No comments

Discussion

Page 11 Line 444-445: I question the claim that women and men enjoy equal access to leisure resources in societies that promote gender equality. Consider research from Australia and other progressive countries and how access to sport and leisure is harder for women and girls due to historical biases, inappropriate facilities etc.

Page 12 Lin 487-489: Was this data collected in the demographics? Either way, you need to state whether this is something you are certain of or speculating about. Similar to my above comment, I think this overlooks a wealth of research on lack of access to sport and leisure for women and girls.

Page 12 Line 513: In the previous paragraph you make a good point about how specific aspects of Turkish culture may influence elements such as societal pressure to work etc., but they you cite a study on a Taiwanese population without noting cultural differences. These need to be noted here as Taiwanese university students will have their own unique cultural barriers and challenges.

Round 2

Reviewer 1 Report

Comments and Suggestions for Authors

Dear Authors,
Your manuscript has been revised to Chapters 1, 4, and 5, but AI-based plagiarism verification shows that generative AI is highly likely. Notably, the similarities between the chapters in question have been confirmed to be high, making it difficult to recognize academic originality in the current state.
Therefore, this manuscript must be rewritten. If the author completely revises the relevant part to ensure originality and then submits it again, we will proceed with a re-examination at that time.
At this stage, it is difficult to proceed with the review.

Author Response

Dear Reviewer 1,

Thank you very much for your careful evaluation of our study. We highly value your sensitivity regarding originality and language quality, and we sincerely appreciate the opportunity to address these concerns.

Our study is the result of nearly seven months of intensive academic work. All sources were accessed directly by us, the data were personally analyzed by the authors, and the contents were written in our own academic style. As English is not our native language, we made use of ChatGPT and Grammarly only for linguistic expression improvements. Beyond this limited support, the theoretical framework, discussions, and literature-based interpretations were entirely produced by the authors.

In addition, earlier draft versions of the manuscript in Turkish also exist. This clearly demonstrates that the originality and the academic content of the study fully belong to the authors.

Moreover, in order to further address your concerns, the relevant sections were submitted to a professional academic proofreading service. Through this process, the text was not only subjected to linguistic expression improvements but also carefully rephrased to strengthen originality and avoid any potential misunderstanding. The official certificate documenting this service has been included in the resubmission package.

In addition, Chapters 1, 4, and 5 have been revised directly by the authors, and explanatory notes were added to indicate the specific changes made in these sections. To ensure transparency, the revised manuscript has also been submitted with the “Track Changes” function enabled, so that all professional language editing modifications can be clearly seen.

In line with the principle of transparency, we also added a statement in the Acknowledgments section of the manuscript clarifying that ChatGPT and Grammarly were used exclusively for linguistic expression improvements.

We believe that these steps will address your concerns and significantly enhance the originality and readability of the manuscript.

Sincerely,
The Authors

Reviewer 2 Report

Comments and Suggestions for Authors

Thank you for your prompt and detailed response to my comments. I am happy to recommend this for publication.

Author Response

Dear Reviewer 2,

We sincerely thank you for your positive evaluation and kind recommendation. We greatly appreciate your valuable time and constructive comments, which helped us to improve the quality of our manuscript.

Sincerely,
The Authors

Round 3

Reviewer 1 Report

Comments and Suggestions for Authors

The authors have adequately addressed all previous concerns, and the manuscript is now clear and well-structured. I recommend acceptance.